# Measured Air Flow Leakage in Facemask Usage

**DOI:** 10.3390/ijerph20032363

**Published:** 2023-01-29

**Authors:** Poul S. Larsen, John Heebøll, Knud Erik Meyer

**Affiliations:** 1Department of Civil & Mechanical Engineering, Technical University of Denmark, 2800 Kgs. Lyngby, Denmark; 2Department of Management Engineering, Technical University of Denmark, 2800 Kgs. Lyngby, Denmark

**Keywords:** flow measurement, filter properties, pressure distribution under facemask, fitter, liner

## Abstract

The importance of wearing a facemask during a pandemic has been widely discussed, and a number of studies have been undertaken to provide evidence of a reduced infectious virus dose because of wearing facemasks. Here, one aspect that has received little attention is the fraction of breathing flow that is not filtered because it passes as leak flow between the mask and face. Its reduction would be beneficial in reducing the dose response. The results of the present study include the filter material pressure loss parameters, pressure distributions under masks, and the fraction of breathing flow leaked versus steady breathing flow in the range of 5 to 30 L min^−1^, for two commonly used facemasks mounted on mannequins, in the usual ‘casual’ way and in a ‘tight’ way by means of three different fitters placed over the mask to improve the seals. For the ‘casual’ mount, leaks were high: 83% to 99% for both masks at both exhalation and inhalation flows. For the ‘tight’ mount with different fitters, the masks showed different lower levels in the range of 18 to 66% of leakage, which, for exhalation, were nearly independent of flow rate, while for inhalation, were decreasing with increasing rates of respiration flows, probably because suction improved the sealing. In practice, masks are worn in a ‘casual’ mount, which would imply that nearly all contagious viruses found in aerosols small enough to follow air streams would be exhaled to and inhaled from the ambient air.

## 1. Introduction

Since the onset of the COVID-19 pandemic, there have been numerous studies on how the spreading of the disease occurs through airborne transmission of aerosols and droplets from human respiration, coughs and sneezes, whether using protective face masks or not [1]. Important aspects include the size of emitted droplets [2], the rate of their evaporation [3,4], and the duration of their suspension and spreading in room air [5]. A key aspect is the protection against the spreading of aerosols/droplets, specifically the transmission of SARS-CoV-2-infected particles. This was studied by [6] who found that cotton, surgical, and N95 masks protected the receiver but that the protective efficiency was higher when the facing spreader also wore a mask. Yet, transmission was not completely blocked, even if masks were completely sealed with adhesive tape. Using the measured concentration of added NaCl aerosol particles, [7] determined the ‘effective filtration efficiency’ (EFE) during inhalation for four types of facemasks mounted on mannequins in a classroom setting. Without mask fitters, EFE for inhalation was low, suggesting leakage rates >50%, while the use of fitters showed EFE values were comparable to those of the ‘material filtration efficiency’ (MFE) and negligible leakage. Other studies, e.g., [8,9], have been concerned with the efficiency of face masks and the need for improved mask design and/or use of mask fitters.

The prevailing view is that face mask filters capture contagious viruses, prevent jets arising to spread the aerosols beyond 1–2 m from mask wearers through breathing, coughing, and talking, etc., retain heavy aerosols and droplets, and supposedly remind people to fulfill various other preventive measures such as keeping social distance, venting rooms, and decontaminating their hands and surfaces, etc. Since early 2020, the importance of an inoculation dose to the morbidity and mortality (dose–response [10]) of the COVID-19 disease has attracted increasing interest, as exemplified by the WHO recommendation that face masks should fit the wearer to reduce unfiltered leakage flows. This problem was pointed out, e.g., by [11], who visualized exhaled leakage flows with smoke studies and thus showed that even simple mouth-and-nose covers made of good filter material cannot reliably protect against droplet infection, because most of the air passes through gaps at the edge of masks. We have come to the same conclusion in a number of visualizations. Such demonstrations, although of qualitative interest, are limited to exhaled flow and cannot elucidate what happens to inhaled flow. Furthermore, they provide no quantitative information on the amount of leakage. A theoretical parameter study by [12] determined the correlation between mask leakage, mask configuration and a person’s facial features, expressed as hydraulic resistance: the ratio of excess pressure under the mask divided by the leak flow. The results, limited to exhalation, show how facial features affect leakage.

Reviewing the literature, we found a lack of data that assessed the efficacy of face mask usage by actually measuring leakage flows. Therefore, we designed an experiment to quantitatively study the fraction of total breathing flow that leaks between the facemask and face during exhalation and inhalation, respectively, at typical flow rates in the range 5–30 L min^−1^ to cover the range of a normal breathing cycle of a mean of 17 L min^−1^ and a maximum of 27 L min^−1^ [13]. We studied two facemasks that were in common use during the 2020–21 pandemic. Masks were mounted on a mannequin head in the usual way, denoted as ‘casual’, but also with three different fitters (liners) mounted on the masks to reduce leak flows, denoted as ‘tight’. Detailed pressure measurements under the masks were used to calculate the filtered flow and hence obtain the %-leakage flow.

## 2. Materials and Methods

### 2.1. Test Facility and Masks

A test facility was designed for the purpose of quantifying flow and differential pressure across masks in simulated human breathing. The assembly consisted of two manifold chambers for pressure measurements (*P*_1_ and *P*_2_, Pa, ±0.5% of reading ±2 Pa by manometers MP120, www.kimu.fr, accessed on 28 October 2022) and an open main chamber with two mannequin heads made from expanded polystyrene (Figure 1). Flow rate from a variac-transformer-regulated air pump (*Q*, L min^−1^) through 13 mm ID steel pipes to or from the mouth of the mannequins was measured by flow meters (MF5700, VB8, Sairgo LTD, Taiwan, ±2%) mounted in the connecting pipes. Two commonly used face masks M1 and M2 (Table 1) were tested without fitters and with two commercial fitters (FTM and Badger), and with one in-house manufactured fitter (GRP) that was molded (of glass-fiber-reinforced plastic) over the mannequin’s face to give a good fit. All tests were carried out at room temperature (18 to 20 °C).

### 2.2. Experimental Procedure

First, to determine the resistance of flow through the filter of a given mask, a sample of the filter material was placed in an adapter with a flow area of *A*_2.5_ = (π/4) 2.5^2^ = 4.909 cm^2^ (Figure 2A). The pressure difference across the filter material Δ*P*_fm_ (Pa) for 4 flow rates, *Q* = 5, 10, 20 and 30 L min^−1^, was used for all tests and was measured with a syringe probe and micro manometer (Furness Control Ltd., FC0510, ±0.25% of reading), plotted and fitted to a linear regression line with slope *c*,
Δ*P*_fm_ = *c Q* = (*K*/*A*_2.5_) *Q*.(1)

In this study, we use the parameter *K* = Δ*P*/(*Q*/*A*) that represents the pressure drop (Pa) per unit flow (L min^−1^) per unit area (cm^2^) of the filter material. This method of determining face mask filter resistance is identical (apart from hardware details) to the standard method [14], which gives results in terms of the so-called standard pressure drop (Δ*P*_std_, Pa cm^−2^) that is related to the present value *K* by the relation,
Δ*P*_std_ = *K* × 8/4.909^2^ = 0.332 *K*.(2)

### 2.3. Determining Leak Flows

Using the array of measured pressure under the mask (Δ*P*_m,i_, *i* = 1 to 28), the flow through the filter (*Q*_filt_) was calculated from
*Q*_filt_ = Σ(Δ*P*_m,i_ × *A*_filt,i_) /*K*,(3)
where *K* is given by Equation (2) and Table 1, and the area at each location was *A*_filt,i_ = 4 cm^2^. The fraction of the total respiratory flow that bypasses the mask as a leakage flow *Q*_leak_ for a given condition was then calculated from
*Q*_leak_/*Q* = 1 − *Q*_filt_/*Q*,(4)
where *Q*_filt_ is given by Equation (3) for the given value of *Q*.

Using a different way to experimentally determine mask leakage, [7] measured the aerosol concentrations *c*_0_ of the ambient air and *c*_2_ of the actually inhaled air, leaving the mannequin wearing a mask. The concentration *c*_2_ arises from the mixing of the filtered flow and leaked flow according to the mass balance, *c*_2_*Q* = *c*_1_*Q*_filt_ + *c*_0_*Q*_leak_, where *c*_1_ is obtained from the efficiency of the filter material. Since *Q* = *Q*_filt_ + *Q*_leak_, the fraction of leakage becomes
*Q*_leak_/*Q* = (*c*_2_/*c*_0_ − *c*_1_/*c*_0_)/(1 − *c*_1_/*c*_0_).(5)

Introducing the measured ‘effective filter efficiency’ EFE = 1 − *c*_2_/*c*_0_ and the ‘effective material efficiency’ MFE = 1 − *c*_1_/*c*_0_ into Equation (5) gives [7] (Equation (9)), which may be written as
*Q*_leak_/*Q* = 1 − EFE/MFE.(6)

To determine the flow through a mask filter, the pressure distribution under the mask was measured over a square, 2 × 2 cm^2^, array at 28 locations by using the syringe pressure probe inserted through the mask filter (Figure 2B–D). Separate tests showed that the observed pressure did not depend on the inclination of the syringe, even though its tip was cut at 30°, implying that the probe recorded the static pressure as the velocity of air under the mask was small except at a point in front of the pipe supplying exhalent flow. Use of the syringe probe left minute holes in the filter material, leading to a reduction in filter resistance *K* versus the number of penetrations per unit area. This quantity was determined by measuring the flow rate and pressure drop over the area of filter material cut from masks M1 and M2 mounted in the adapter, subject to the number of syringe penetrations *n* = 0, 10, 20, ..., 50.

## 3. Results

Measured values of pressure drop versus flow through mask filter samples mounted in the adapter showed highly linear correlations (R^2^ = 0.995–0.997), giving the *c*-values (Equation (1)) and converted values (*K* and Δ*P*_std_) listed in Table 1. The resistance to flow (in units of Pa cm^2^ per L min^−1^) through the M2 filter (*K* = 206) was twice that through the M1 filter (*K* = 102). The change in *K*-value of M1 and M2 filters for the increasing number of perforations in the filter material made by the syringe probe was determined to be –0.187 and −0.795 Pa cm^2^ per L min^−1^ per hole, respectively, in the small area *A* (= 4.909 cm^2^) of the adapter. These results imply systematic errors of 0.23% and 0.51%, respectively, for the *K*-values of the M1 and M2 mask areas (110 cm^2^) after the first 28 perforations, or about 1% and 2% at the end of four pressure distributions at flows of 5, 10, 20 and 30 L min^−1^ for the given mask study.

Figure 3 shows all results for %-leakage flow (Equation (4)) versus simulated steady respiration flows of exhalation and inhalation from 5 to 30 L min^−1^ for masks M1 and M2 in ‘casual’ mounts and ‘tight’ mounts of different fitters, respectively. In the ‘casual’ mount, leakage is systematically high, 83 to 99%, while for the ‘tight’ mount, it varies from 18 to 66%, depending on the fitter. The case of the M1 ‘casual’ mount for exhalation was repeated (Figure 3A □), and the single point of *Q* = 30 L min^−1^ for M1 ‘casual’ at exhalation and inhalation was also repeated (Figure 3A,B ∆). Examples of measured pressure distributions (Figure A1 in the Appendix A) show regions of nearly uniform high pressure around the mouth area, surrounded by regions of decreasing pressure towards the edge of the mask where the gauge pressure becomes zero.

## 4. Discussion

It is known that the use of protective face masks is beneficial in reducing the spread of contagious viral diseases, but also that not all breathing air is being filtered, owing to leaks between the mask and face. This is readily visualized by smoke studies for exhaling flows (e.g., [11]). Leaks could also be expected for inhaling flows but this cannot be visualized through similar means. In fact, the present experimental results (Figure 3) show that the %-leakage flows are high (83% to 99%) and almost the same for exhalation and inhalation for the two masks, M1 and M2, in a ‘casual’ mount. A repeated measurement for a ‘casual’ case may give a different result (Figure 3A,B □ ∆), because the casual mounting of a face mask may vary from time to time. However, this is not the case for a ‘tight’ mount where the fitter ensures more repeatability (Figure 3A × ●).

For exhalant respiration flow (Figure 3A), the three fitters in the M1 ‘tight’ cases show considerable differences in reducing leak flow. Nearly independent of the rate of flow, our measurements show the leak rates: FTM (~60%), Badger (~50%), and GRP (~25%). The in-house manufactured GRP fitter was molded over the mannequin’s face to give a good fit here, as the two commercial fitters have adjustable features that may not give optimal fit for all faces. The average leakage flow of a mask with a given fitter (except the Badger) is nearly the same for exhalation and inhalation, as is also observed by [15], but the magnitude depends on the respiration rate. Thus, it is noted that increasing rates of inhalant respiration flow (Figure 3B) lead to strongly decreasing leakage, where our measurements show FTM (~55% mean leak), Badger and GRP (~25% mean leak). The decreasing trend suggests improved sealing due to increasing suction under the mask for increasing inhalant flow. For exhaling flows, one should expect an opposite effect of increasing leaks for increasing flow and pressure under the mask, but this is only seen for the M2 ‘tight’ mount (Figure 3C), probably because the resistance to filtered flow of M2 is twice that of M1 (unless the first data point at 5 L min^−1^ is an outlier).

We may estimate the accuracy of the present results by noting that the error of each measurement of pressure under the mask may be as high as ±0.5 Pa. For ‘casual’ cases of high leakage, pressures are typically low, only a few Pa, and, for a conservative estimate, assuming a one-sided increase of 0.5 Pa to all measured values over the mask array for M1 ‘casual’ inhalation, for example, we obtained errors of 2.5 to 1.4% leakage for the flow range of *Q* = 5 to 30 L min^−1^. For comparison, for inhalation with the M1 ‘tight’ mount with the GRP fitter, pressures are high, up to 14 Pa at *Q* = 10 L min^−1^ and 60 Pa at *Q* = 30 L min^−1^, filter flows are high, and leakage is low. Therefore, a systematic one-sided error of 0.5 Pa on all measurements would here give the higher 6.3 to 1.4 % leakage for the flow range. Another source of error of the present results could stem from the 28 unit areas, *A* = 4 cm^2^, marked on masks for pressure measurements. A sensitivity calculation for the M1 mask ‘casual’ and ‘tight’ mounts with the GRP fitter during inhalation at 10 L min^−1^ shows that a 1% change of all areas would give a 0.6 and 3% change, respectively, in leakage rate. A reduction in areas, say due to some folding of the mask, would imply an increased leakage flow, while an unlikely underestimate of the areas would imply a decrease in the leakage flow. However, such errors would not close the gap to the results of [7], as discussed next.

The present magnitudes of leakage during inhalation may be compared to those derived from concentration measurements obtained by [7]. They state that MFE ~0.98 for the ~0.1 µm aerosol particles used and their Table 1 gives EFE = 0.447 and 0.913 for the surgical mask with and without a Badger fitter, respectively. Using Equation (6), this gives about 55% and 8.7% leakage, respectively, compared to the present M1 values of 95% for ‘casual’ and 30–50% for ‘tight’ with a Badger fitter. The results assume that particles as small as indicated will closely follow the leak flow that bypasses the mask filter, but be fully retained by the filter material and not deposited on other surfaces. Low values of *c*_2_ in Equation (6) lead to high values of EFE and low values of leak flow. The differences in %-leakage flow between the results in [7] and those of the present study could therefore be related to the process of particle deposition, which has not been considered. While we only consider the airflows, there remains the important question of the amount of aerosolized pathogens that are actually inhaled from the ambient or expelled to the ambient via leakage flows, because of a possible reduction in concentration due to the deposition of such particles. This problem would require a different experimental and/or theoretical study of the process of impact and adherence and hence the deposition of such particles on facial surfaces swept by leakage flows.

## 5. Conclusions

Filter resistance to flow in masks is a compromise between the ability to retain aerosol particles and the ability to prevent leakage, because flow through the filter surface is proportional to the pressure difference across the filter. For ‘tight’ mask mounts, seals seem to be sufficiently good to allow the buildup of pressure to force 40% to 80% of the flow through filters, while the remainder leaves as leakage flow (Figure 3). For ‘casual’ mask mounts, however, it appears from all cases tested that the seals are insufficient to allow for increasing pressure differences across the filter; hence, almost all flow bypasses the filter as leak flow (Figure 3 ○ □ ∆). It may, therefore, be concluded that, since face masks in practice are often worn in a ‘casual’ mount, nearly all contagious viruses found in aerosols that are small enough to essentially follow the air streams will not be retained by the masks, but exhaled to the ambient and inhaled from the ambient. Here, the benefit of masks is associated with the location of contagious regions, now being confined to the sides of the face rather than ahead of the face. Nevertheless, any reduction in leaks would improve protection and reduce the virus dose at inoculation. Face mask wearers should be aware of this fact and mount masks with care to reduce the openings between mask and face, or use a fitter.

Combined with the dose–response impact on the morbidity and mortality of the SARS-CoV-2 virus [10], the present study supports the significance of obtaining the better sealing of face masks. A way of perceiving the importance is to compare masks with vaccines: they do not prevent infection but they alleviate symptoms and disease. This statement is valid for any aerosol-borne viral disease where the dose–response is of essence.

## Figures and Tables

**Figure 1 ijerph-20-02363-f001:**
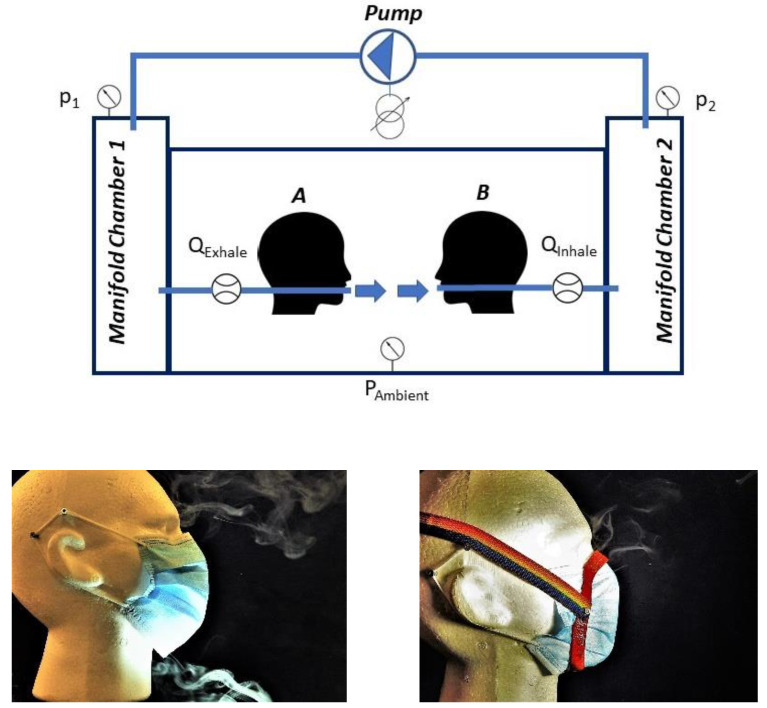
Upper: test facility with mannequin heads for exhalation (A) and inhalation (B) on which masks are mounted. Air flow rate was regulated by a variac transformer driving an air pump and measured by flow meters placed in pipes between chambers and mouth regions, and chamber pressures measured by pressure manometers. Lower: surgical face mask M1 mounted on mannequin’s face ‘casual’ (**left**) and ‘tight’ by use of the GRP fitter (**right**) where leak flows visualized by e-cigarette smoke show chin and nose leaks (**left**) and nose leak (**right**).

**Figure 2 ijerph-20-02363-f002:**
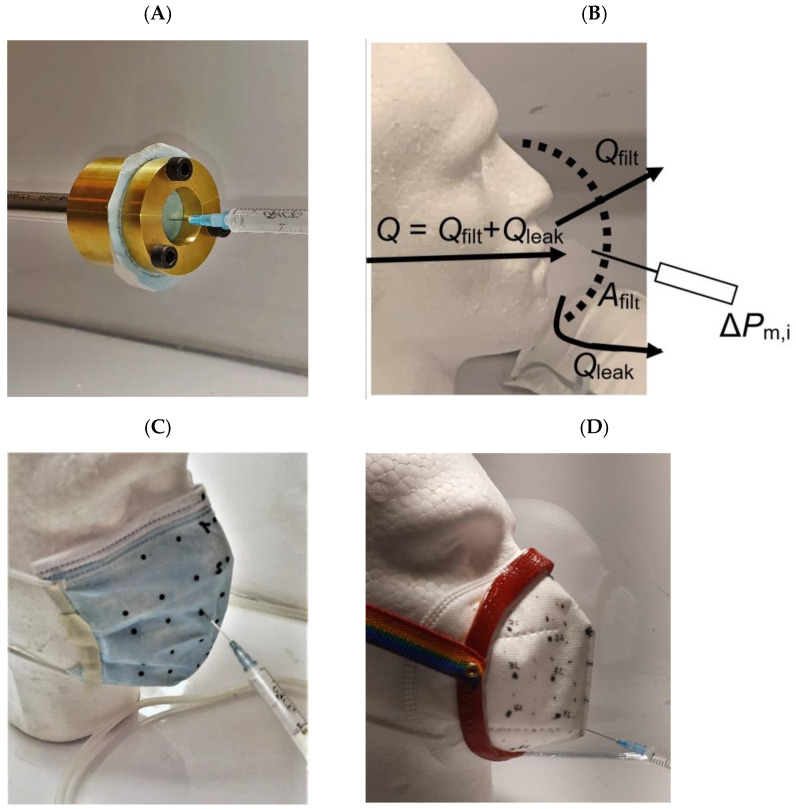
(**A**): Adapter with filter sample and syringe pressure probe. (**B**): Definition of filter and leak flows showing schematic use of pressure probe inserted through the filter material to measure the local pressure Δ*P*_m,i_ under the mask. (**C**): M1 mask in ‘casual’ mount showing marked array of 28 locations for measuring local pressure distribution under the mask. (**D**): M2 mask in ‘tight’ mount with GRP fitter showing marked array and pressure probe.

**Figure 3 ijerph-20-02363-f003:**
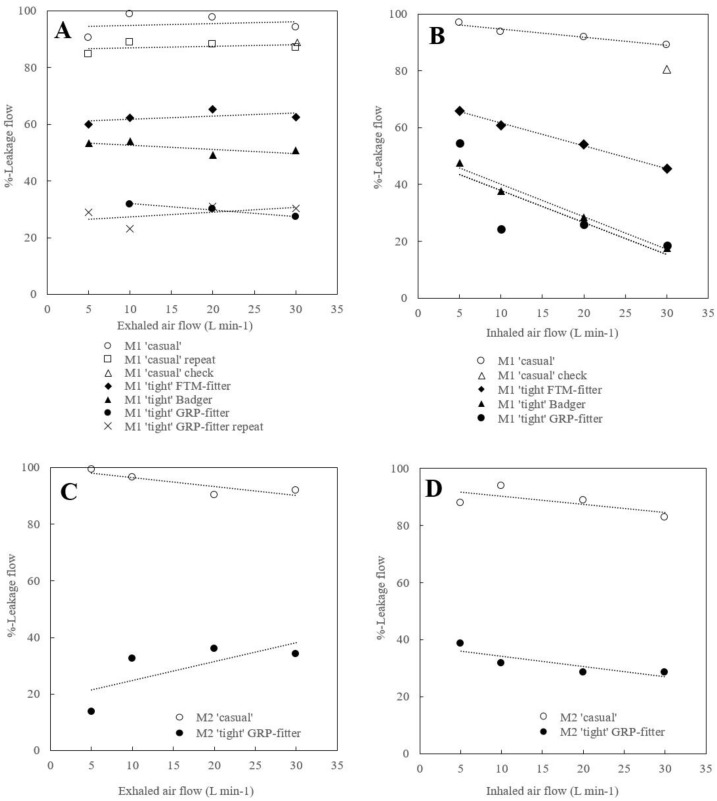
Percent of leakage flow *Q*_leak_/*Q* versus simulated steady respiration at exhalation (**A**,**C**) and inhalation (**B**,**D**) for masks M1 (**A**,**B**) and M2 (**C**,**D**) in ‘casual’ and ‘tight’ mounting by fitters, respectively. Trends are illustrated by the linear regression lines.

**Table 1 ijerph-20-02363-t001:** Face masks and filter properties. Measured resistance *c* (Pa per L min^−1^ through area 4.909 cm^2^), ‘standard resistance’ Δ*P*_st_ (Pa cm^–2^, see text), *K* (Pa cm^2^ per L min^−1^), and filter area *A*_filt_ (cm^2^). The ‘popular surgical mask’ is similar to what is known as the ‘medical procedure mask’.

Face Mask	Description	*c*	Δ*P*_st_	*K*	*A* _filt_
M1	Protego (TKK) Surgical mask, EN 14683 Type I, Model TNK-KZ 3-layer, disposable, widely used by the public	20.8	33.9	102	110
M2	Weifei 9520 EN149:2001 + A1:2009, FFP2 NR 5-layers, non-woven, Melt-blown, Cotton	42.0	68.4	206	110

## Data Availability

All data are available from the corresponding author upon request.

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
