# Peer review of "Measured Air Flow Leakage in Facemask Usage"

_ijerph, 2023, doi:10.3390/ijerph20032363_

Round 1

Reviewer 1 Report

Comments to Measured air flow leakage in facemask usage

 Through the experimental device designed, the authors quantitatively studied the relationship of the leakage rate of M1 and M2 masks under different wearing modes ("casual" and "title") with the total flow rate. The manuscript is well written and the results shows the significance of better sealing of masks. However, I have a couple of questions for the manuscript, and the authors should revise the manuscript for publication in Environmental Research and Public Health. Here are some comments:

1.It is recommended to provide more detailed data on these two masks. For example, mask body structure and mask size, etc.

2.Please put the four parts (A,B,C and D) of Figure 2 on one page for reading easily.

3. Page 6. The clarity of Figure 3 should be improved. Please check it.

4.The %-leakage flow on line 173 is 85-99%?not 82-99%?

5.Page7, line183: The author say that for the same fitter, the leakage flow of exhalation and inhalation is almost the same, however, as can be seen in Figure 3A and 3B, there is a certain difference between the average leakage rate of exhalation and inhalation with the FTM fitter. Please check it.

6. Page7, line186: The author mentioned that the leakage flow is related to the total flow of the inhalation, the greater the total flow, the lower the leakage flow. However, figure 3 does not show the regression line for the GRP fitter being used. Which shape pattern represents the GRP fitter? Please indicate in the figure note.

7. Page7, line194: The author mentioned that the error of each measurement of pressure under the mask may be as high as ±0.5 Pa, and for a conservative estimate assuming a one-sided increase of 0.5 Pa to all measured values for M1 'casual' inhalation, according to what?

8. Page7, line217: The authors say it is difficult to explain the difference in the leakage flow calculated by these two methods.

First, the authors say that particles as small as indicated will closely follow the leak flow that bypasses the mask filter, but be fully retained by the filter material and probably not deposited on other surfaces, this is an important analysis, and are there any other reasons for this phenomenon? Whether the M1 mask used in this study and the surgical mask used in literature [7] have the same filtration efficiency for 0.1 μ m particles?All of these reasons will lead to different C2, which leads to a large difference in the leakage rate calculated by the two methods. Therefore, in order to avoid factors that cannot be considered controlled, if authors would like to study the differences between the two methods, it is necessary to test the MFE value and EFE value of the M1 mask used in this study, and then compare the two methods to analyze the reasons for the difference between the two methods.

9. The authors emphasize the importance of sealing of mask, this study shows that fitter can significantly reduce the leakage flow, so whether the author recommends that the public masks be used with the fitter?

10.Please separate the conclusion and discussion in Section 4, conclusion as a separate section, and results and discussion can be placed in a section or independently.

Reviewer 2 Report

The presented article discusses interesting and innovative research, but these are laboratory analyzes of the course of the human breathing process through a protective mask, which describe the problem of the accuracy of the mask's adherence to the face. However, they do not take into account many factors that can improve or worsen the accuracy of mask adhesion, such as the use of makeup or facial hair. The article also does not take into account the cyclicality of inhalation and exhalation, which in the case of increased tightness increases the amount of air returned back to the lungs, which leads to a decrease in the amount of oxygen in the inhaled air. Also, no reference was made to the dependence associated with the increase in air flow resistance through the mask on lung capacity, which is especially important in people with respiratory problems. Perhaps the research should be extended to include parameters not taken into account so that they can bring new elements in a given field of science.

The presented drawings and tables are legible and show the performed calculations. The conclusions are consistent with the evidence presented. The cited articles are also correct.

Round 2
